# TI-P: Tactile-based Interactive Motion Planner in Unknow Cluttered Environemnts

Chengjin Wang
*Shanghai Research Institute for Intelligent Autonomous Systems*
*Tongji*
*University*
Shanghai, China
2210991@tongji.edu.cn

Yan Zheng
*Shanghai Research Institute for Intelligent Autonomous Systems*
*Tongji*
*University*
Shanghai, China
2310292@tongji.edu.cn

Yanmin Zhou*
*College of Electronics and Information Engineering*
*Tongji*
*University*
Shanghai, China
yanmin.zhou@tongji.edu.cn

Zhipeng Wang
*College of Electronics and Information Engineering*
*Tongji*
*University*
Shanghai, China
wangzhipeng@tongji.edu.cn

Runjie Shen
*College of Electronics and Information Engineering*
*Tongji*
*University*
Shanghai, China
11132@tongji.edu.cn

Bin He
*College of Electronics and Information Engineering*
*Tongji*
*University*
Shanghai, China
hebin@tongji.edu.cn

*Abstract*—**Robotic motion in unknown cluttered environments often failures from catastrophic collisions and obstructions due to constrained free-motion space and light-suffer challenges. Multimodal tactile perception with force and proximity sensing offers inherent advantages in overcoming these limitations. This paper proposes a tactile-based interactive motion planning method (TI-P) using multimodal tactile sensing, which utilizes real-time tactile feedback to perceive the environment and infer the force-displacement characteristics of interacting objects. These interaction features are integrated into a sampling-based motion planner to predict the maximum connectivity probability of candidate trajectories. Subsequently, the planner interpolates sampled points and extrapolates the motion of objects along the trajectory to compute the optimal interaction forces for driving the robot. Simulation results demonstrate that the proposed planner effectively guides the robot to compliantly manipulate obstacles in its path, significantly improving motion adaptability in unknown cluttered environments.**

*Keywords—interactive motion planning, tactile perception, unknown cluttered environments, perception-motion closed loop*

## I. INTRODUCTION

The recent surge in research interest surrounding the autonomous motion of robotic agents in cluttered environments stems from their potential applicability in community-level scenarios [1-4]. These applications span unstructured domains such as households[5] and elderly care facilities[6], where environmental unpredictability necessitates superior motion adaptability compared to structured industrial environments.

The primary objective of deploying robots in community environments is replace human labor, mitigate workforce shortages, and enhance daily convenience[5, 7]. However, such environments often feature highly cluttered spaces due to efficient space utilization and human living habits. This results in light-suffer and constrained free-motion space, posing significant challenges to robotic perception and movements. While visual perception remains prone to high uncertainty, catastrophic collisions may lead to motion failure. In contrast, humans rely on tactile perception to perceive their surroundings and reconfigure the spatial state of manipulable objects to facilitate movement—a capability that remains challenging for robots to replicate.

Prior studies have incorporated tactile feedback at the control level to achieve compliant environmental interaction through contact force regulation [2, 8]. However, the success of such methods heavily depends on predefined motion trajectories. Recent advances explore tactile-aware motion planning to enhance robotic adaptability in cluttered scenes [9-11]. For instance, [9] introduced a movement primitive-based planning method, where tactile signals are mapped to predefined motion primitives for tactile-guided navigation. In environments with movable objects, a physics simulation-aided planner is proposed[11]. The pre-optimizes actions in simulation can prevent catastrophic collisions. Nevertheless, these methods rely on prior knowledge of object interaction properties, limiting their deployment in unknown environments. Ideally, robots should autonomously infer interaction characteristics and integrate such knowledge into planners to generate adaptive interaction strategies.

Inspired by human tactile-guided interaction behaviors, we propose a Tactile-based Interactive Motion Planner (TI-P)—a closed-loop framework constrained by multi-dimensional object interaction features. The TI-P architecture comprises:

1) An environment understanding module that infers object interaction features from multimodal tactile data

generated during bodily interactions, enabling behavior-guided perception;

2) A planner module that generates interpretable interaction actions using these features, achieving perception-guided behavior.

By integrating real-time tactile inference with spatial state reconfiguration of operational objects, TI-P actively expands the free motion space, significantly enhancing robotic adaptability in unknown cluttered environments (Fig.1).

The planner employs a sampling-based method to compute intermediate waypoints with maximal connectivity probability, using operational-weighted grid maps and target positions as constraints [12]. These waypoints are interpolated to generate a reference trajectory. For execution, an impedance controller tracks the trajectory while maintaining compliant interaction [13] (Fig.2).

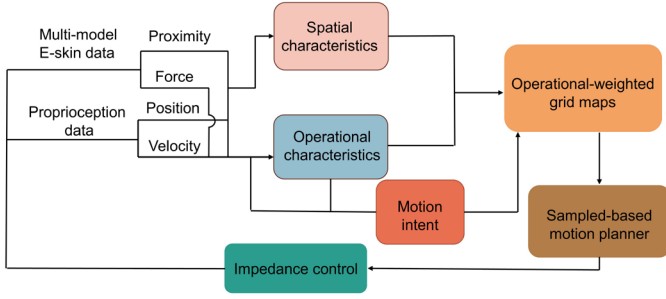

Fig.1: Overview of Tactile-based Interactive Motion Planner (TI-P).

We constructed a cluttered tabletop environment in PyBullet [14] to evaluate TI-P, simulating real-world community settings. The scene contains cylindrical objects with randomized physical properties, fixed at arbitrary locations. During testing, the workflow begins by generating a random target pose within the workspace with number range of [1, 4]. And then loading six objects occupying 57% of the workspace volume. Ten trials were conducted for each test condition.

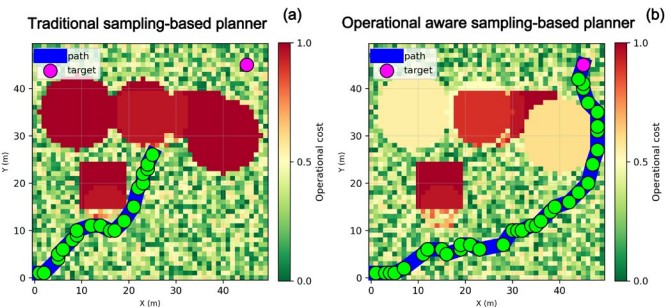

Fig.2: Example comparing the performance of sampling-based planners under constrained free-motion workspace. (a) A boundary-constrained sampling-based planner fails to find a feasible path due to the absence of collision-free solutions. (b) An operational-feature-constrained sampling-based planner successfully completes the task in the same constrained workspace by inferring object interaction characteristics to generate interactive trajectories.

We benchmarked TI-P against a boundary-constrained sampling-based planner (BS-P) with impedance control.

Experimental results demonstrate that TI-P achieves a 55% higher success rate, attributed to two key advantages:

1. The baseline fails when intermediate arm links are blocked by fixed objects. TI-P circumvents this by dynamically delineating restricted zones in the configuration space.

2. The baseline's impedance control generates insufficient interaction forces to displace movable objects. TI-P overcomes this by applying force compensation based on real-time interaction characteristics $\theta$ identification.

TABLE I. PERFORMANCE EVALUATION AMOBG TI-P AND BS-P

| Planning Methods | Number of target points | | | |
|---|---|---|---|---|
| | 1 | 2 | 3 | 4 |
| TI-P | 100% | 100% | 80% | 60% |
| BS-P | 30% | 10% | 0% | 0% |

Furthermore, we tested the motion performance of TI-P in a cluttered and crowded cabinet scenario. This environment contains a large number of movable and immovable everyday objects. These items are tightly arranged inside the cabinet, making it difficult for the robot to find a collision-free path. The objective of the experiment was to have the robotic arm reach and touch a beverage located deep inside the cabinet.

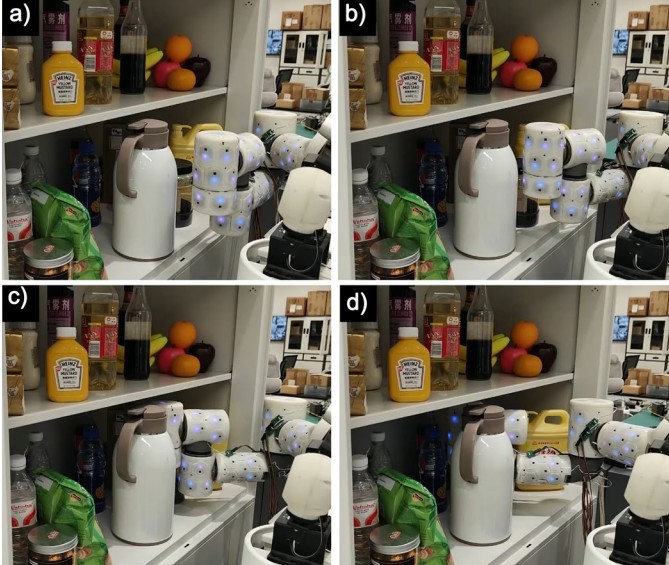

Fig. 3: Sequence of the robot touching the beverage inside the cabinet.

The test results demonstrate that TI-P can reliably update the manipulation cost of objects in the environment through tactile perception data. These updated costs are then used to constrain the sampling-based motion planning method. When the robot made contact with the kettle, tactile exploration led to its manipulation cost being updated to 1. Since this exceeded the maximum manipulation capability (Fig. 3a), a bypass strategy was adopted (Fig. 3b). Subsequently, upon contact with the plastic bottle and given its lower manipulation cost, the robot expanded the free motion space by repositioning the bottle (Fig. 3c), ultimately achieving contact with the beverage (Fig. 3d).

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
