# OpenReview forum: "TI-P: Tactile-based Interactive Motion Planner in Unknow Cluttered Environemnts"
_IEEE.org/IROS/2025/Workshop/Tactile_Sensing — IROS 2025 Workshop Tactile Sensing Poster_

### Official Review · Reviewer_rC2r · 2025-09-21
**Motion planner with touch perception**

**Rating:** 8
**Confidence:** 4

**Review:**

This paper proposed a method to integrating touch-sensing into sampling-based motion planning. The high-level idea is to first infer object characteristics (i.e. its compliance, modeled by a spring-mass-damper system) from the touch information. Then, a planner is used to select a 'safest' / 'easiest' path towards the goal (the metric is defined by object characteristics).

Suggestion: It would be great to have a figure to demonstrate a sequence of "touch exploration" and show how the planner's cost function is updated accordingly.

---

### Official Review · Reviewer_GxRW · 2025-09-22
**A promising framework combining tactile sensing and planning, though the experimental evaluation is limited.”**

**Rating:** 6
**Confidence:** 4

**Review:**

This paper proposes a tactile-based interactive motion planner (TI-P) for enabling autonomous robot motion in cluttered environments. The system integrates multimodal sensory inputs—including proximity sensing, force feedback, electronic skin (e-skin), and proprioceptive data (joint positions/velocities)—with a modeling approach for interaction characteristics.

The work presents an interesting direction by combining tactile property estimation with motion planning and demonstrates preliminary effectiveness in simulation. To further strengthen the contribution and increase the credibility of the results, I would encourage the authors to consider the following improvements:

(1) Incorporating real-world robotic experiments to validate the approach beyond simulation,

(2) Benchmarking against a broader set of baselines,

(3) Providing sensitivity or ablation studies to better highlight the impact of different components.

Overall, the paper has potential, but these additions would significantly enhance its quality and persuasiveness.